# Why Do Consumers Intend to Purchase Natural Food? Integrating Theory of Planned Behavior, Value-Belief-Norm Theory, and Trust

**DOI:** 10.3390/nu13061904

**Published:** 2021-06-01

**Authors:** Valentina Carfora, Carla Cavallo, Patrizia Catellani, Teresa Del Giudice, Gianni Cicia

**Affiliations:** 1Department of Psychology, Catholic University of the Sacred Heart, Largo Agostino Gemelli, 1, 20133 Milan, Italy; valentina.carfora@unicatt.it (V.C.); patrizia.catellani@unicatt.it (P.C.); 2Department of Agricultural Sciences, University of Naples Federico II, Via Università 100, 80055 Portici, Italy; agriqual@unina.it (T.D.G.); cicia@unina.it (G.C.)

**Keywords:** natural food, theory of planned behavior, value-belief-norm, purchasing intention, pro-environmental motives

## Abstract

Natural labels are increasingly present in the market and appreciated by consumers, despite formal regulation still missing. Knowing the psychosocial factors that may predict natural food choice may be useful to understand what drives consumers to choose this category of food. We analyzed the antecedents of consumers’ intention to purchase natural food, testing a theoretical model that integrates the theory of planned behavior (TPB), the value-belief-norm (VBN) theory, and consumers’ trust in natural food. A sample of Italian participants (*N* = 1018) filled an online questionnaire assessing intention to buy natural food, TPB and VBN variables, and trust in the natural food supply chain. The model applied yielded results which confirmed the predictiveness of the tested integrated model. Attitude and perceived behavioral control were the strongest antecedents of intention, followed by trust and personal norm. Consumers’ intention to buy natural food was also associated with their evaluation of the consequences and possibilities related to the purchase behavior, as well as with their moral evaluation attributable to pro-environmental determinants.

## 1. Introduction

Natural-labeled food permeated food retailers in several Western countries, such as US, where it is deemed to be the leading food label, especially for new or renewed food products [1,2]. Their presence has been estimated to account for 25% of new foods [3,4,5].

Despite the increasing success, we still miss an uncontroversial definition of natural food. According to European Union guidelines, the *natural* claim is allowed when an ingredient is naturally contained in that food, or when the product has no chemical additives [Regulation (EC) No. 1924/2006, Regulation (EC) No. 1047/2012]. In the US, the Department of Agriculture (USDA) stipulated that, to define a food as “natural”, it must have no artificial flavoring and be minimally processed [6]. However, this definition is not exhaustive, as most of the food currently sold has undergone some sort of processing and no explicit procedure has to be followed to obtain this label [7]. The confusing official definition had backlashes to the point that several legal lawsuits and class actions were initiated for inaccurate *natural* labeling [8]. Nevertheless, this labeling it is generally considered appropriate when artificial additives have not been used, even if consumers can attach to it every meaning they wish [1]. Natural labels can also overlap in their meaning with other claims, such as organic and the so-called *clean* labels: They can be “free from” labels of reformulated products with the absence of ingredients that are perceived to be somewhat harmful [9]. As a matter of fact, we underlined some confusion on the consumers’ perception side, but, instead, the organic labeling is strictly normed. So, *organic* food carries a specific label (that can be identified with the green leaf logo) and has to satisfy specific production requirements as indicated by the law [Regulation (EC) n. 834/2007]. Here, the case study is represented by food with a claim or label that contains the *natural* word, that is not certified organic, also excluding products with a “free from” label. 

Starting from Rozin (2005) [10] and Rozin et al. (2004), there have been several studies devoted to understanding rational and moral motives that might drive consumer intention to purchase natural labels. Regarding the rational motives, previous scholars have showed that a substantial health halo surrounds natural products, and this may lead consumers to misbehave and worsen their health, instead [11,12,13,14]. The heuristic linking natural to good seems to be quite powerful and long-lasting. For example, an experiment by Li and Chapman (2012) showed that consumers do not believe researchers telling them that some foods with or without the natural label have exactly the same ingredients. Furthermore, consumers associate natural food with an improved taste and greater freshness [4,7,11]. In addition, natural food is perceived as healthier than processed food. A popular belief is the one according to which food processed by humans is harmful [13]. Indeed, processed foods are perceived riskier, although they are the result of the evolution of food as we know it nowadays: high-quality in terms of sensory characteristics, shelf life, nutritional profile, etc. [15,16,17]. Actually, Rahman et al. (2020) [18] reported that the majority of consumers are worried that processed food can be more noxious than fresh food. Perceived risks originated in food scandals of past years and in the fear towards unknown ingredients as Genetically Modified Organisms (GMOs) [19,20]. Finally, natural foods have a short ingredients list, that is easy to process and understand even in a quick shopping occasion—as happens for most of foods [21,22]. As to the moral and pro-environmental motives that might drive consumers’ intention to purchase natural food, consumers associate natural food with higher respect for the environment and animal welfare [15]. These ideas are independent on the features actually embedded in these products, leading to a proper *natural bias* [1,7]. The perception of naturalness is mainly linked to a pastoral view of nature and natural entities, especially in urban contexts, where naturalness is seen as detached from tradition and compatible with innovation [3]. 

Despite the value of the above studies, our current understanding of the complex relationships among the diverse psychosocial motives associated with natural food purchase is still far from being solid. So far, scholars seem to never have considered rational and moral motives simultaneously nor have analyzed if the consumers’ trust in the natural food supply chain towards natural food has a role in determining its purchasing. In the present study, we contribute to this debate, integrating rational motives, pro-environmental motives, and trust in a unique theoretical framework. We therefore propose an integrated theoretical model to explain consumers’ intention to purchase natural food, and teste it through a nationally representative survey that has been carried out in Italy. Our contribution is aimed at being a first step towards a theoretical model that suitably accounts for the complexity of the consumers’ attention to diverse aspects of natural food products. 

## 2. Theoretical Background

In the current literature, no scholars have precisely focused on the psychosocial predictors of consumers’ intention to purchase natural food. Thus, we based our study on the literature on consumers’ intention related to sustainable food consumption, such as local food products [23], fair-trade food products [24], and organic foods [25]. Within this literature, scholars have analyzed the role of different psychosocial antecedents of consumers’ intention and behaviors, such as food sustainability and environmental concerns [14,26], availability in grocery stores nearby family homes [27], perception of obstacles to sustainable food products [28], consumers’ attitude [29,30], and consumers’ trust [31]. Accordingly, a meta-analysis reported consumers’ attitude, subjective norm, perceived behavioral control, beliefs, personal norm, and ethical concerns as main predictors of sustainable food consumption [32]. Note that these predictors are key variables of the Theory of Planned Behavior [33] and the Stern’s Value-Belief-Norms theory [34]. The TPB is a self-interest-based explanation of consumers’ choices, with a notable criticism related to the omission of a moral effect consideration. Contrarily, the VBN focuses on the predictive roles of values and moral norms, but neglects to assess cognition and reasoning on particular issues [35]. Although these two models are usually employed separately, some scholars have tried to integrate them to explain different consumers’ intentions [36,37,38,39,40,41,42,43]. For instance, Chen (2020) recently confirmed that the integration of both models well predicts people’s intention to consume locally produced organic food. However, all the previous studies on the TPB and VBN integration to predict the purchase of food products lacks the consideration of a key predictor of people’s intention to buy foods, that is trust in the food supply chain. Trust in the supply chain is an emerging topic in literature, even if its role has greatly been considered using qualitative rather than quantitative tools [44]. Importantly, there are still few studies that investigate its predictiveness within integrated psychosocial models aimed to explain consumers’ intention to buy food products. Following this, in the present research, we aim at explaining consumers’ intentions to purchase natural food by integrating VBN and TPB models plus consumers’ trust in the natural food supply chain. The two models and the additional role of trust are discussed below. 

### 2.1. The Theory of Planned Behavior

The TPB [33] is a theory that explains the psychological determinants of decision-making, which posits that people make reasoned decisions to engage in a specific behavior. The TPB is frequently applied to the study of consumer food choice [32,45]. According to the TPB, consumer behavior is predicted by behavioral intention, which reflects motivations and cognitive planning for engaging in such behavior. In turn, intention is determined by attitude, subjective norm, and perceived behavioral control. Attitude is related to the evaluation of the positive and negative consequences associated with the behavior. Subjective norm refers to the perception of social expectations towards the behavior. Finally, perceived behavioral control is attributable to the individual evaluation of being able to engage in the behavior [33]. 

Prior studies on food choice showed that attitude and perceived behavioral control are important antecedents of people’s intention, while results regarding the predictive power of subjective norm are less consistent [46,47,48,49]. On the one hand, according to some meta-analyses, subjective norm is the weakest antecedents of intention [50,51]. On the other hand, many empirical studies demonstrated its role in explaining intention [52]. Specifically, in the case of the Italian consumers, Vassallo et al. [28] found that perceived behavioral control was the strongest predictors of sustainable food choices.

### 2.2. The Value-Belief-Norm Model

One of the main critiques of the TPB has been to overlook people’s moral and normative drives [50,53]. Moral norm is related to the people’s consideration of the moral rightness of a behavior [54], and several studies have supported its inclusion to increase the explained variance of intention [55,56], especially in the case of sustainable food choices [57,58]. To analyze how moral factors determine consumers’ intentions and behaviors in the case of sustainable food choices, in this paper, we refer to the VBN theory, which argues that when an individual is aware of the outcomes of her behaviors and is willing to assume responsibility for these outcomes, moral norm is activated. The operationalization of this theory consists of measuring behavioral intention as determined by the personal norm, which is the consciousness of an obligation to behave according to our moral principles. Personal norm, in turn, is explained by the ascription of responsibility, that is an individual feeling of being responsible towards the environment. Moreover, ascription of responsibility is determined by the awareness of the outcomes of human actions, which in turn derives from general pro-environmental beliefs [59]. 

Within the literature on the antecedents of food choice, the VBN variables have already been shown to explain sustainable food choices [57,60,61,62]. For instance, a strong endorsement of self-transcendence values leads to a lower intention to eat meat [63], while the awareness of consequences determines a higher ascription of responsibility, and then a higher personal norm towards the preference of an organic menu (Shin et al., 2018) [64]. Similarly, biosphere values, environmental beliefs and norms are believed to guide environmentally conscious food choices among college students [65].

### 2.3. The Additional Role of Trust

In the present paper, we also verify the predictive role of trust in explaining consumers’ intentions to purchase natural food. Trust plays a relevant role in the decision-making process during the purchase of food products [66], given that only a few consumers are aware of the background of food production and most of them cannot verify it [67]. Thus, consumers need to trust that the food product they are going to buy is authentic and genuine [68]. Consumers’ trust is specifically pertinent in the current study, because consumers generally have little information about natural food, as well they receive little exposure to the production or preparation of this category of food [69]. Within the TPB application to predict consumers’ choice of food products, recent studies have confirmed that trust in the food supply chain is a relevant precursor of consumers’ purchase intention [70].

### 2.4. Objectives and Hypotheses

In the present study, we aimed to predict consumers’ intention to buy natural food by evaluating the predictiveness of both rational and moral considerations. Therefore, we tested the integration of TPB and VBN theories, also including a measure of trust in the natural food supply chain. Our hypotheses are described below and illustrated in Figure 1.

Consistent with the TPB model, we hypothesize that participants’ intention to purchase natural food is predicted by their attitude towards natural foods (H1), subjective norm regarding purchasing natural food (H2), and perceived behavioral control (PBC) (H3). 

Consistent with the VBN model, we also expect that participants’ intention is determined by the sequential chain of personal norm (H4), ascription of responsibility (H5), awareness of consequences of purchasing natural food (H6), pro-environmental beliefs (H7a). To test the predictiveness of the chain of paths proposed by the VBN model, we also verify whether general pro-environmental beliefs have an indirect impact on intention via the sequential mediation of the other beliefs: awareness of consequences, ascription of responsibility, and personal norm (H7b). 

In our study, we also test a series of relationships among VBN variables, TPB variables and trust, formulating six additional hypotheses. Within the TPB model, attitude is related to individual beliefs regarding the outcomes of undertaking a specific behavior as a function of the personal evaluation of the consequences. Thus, we expect that consumers’ attitude to purchase natural foods is predicted by consumers’ general pro-environmental beliefs (H8) and awareness of the consequences (H9).

Furthermore, a number of scholars confirmed the hypothesis that voluntary control is an antecedent of moral responsibility [71,72]. However, so far, none of the studies on food choices analyzed whether the consumers’ perceived behavioral control predicts ascription of responsibility. Thus, in the present study, we also verify whether consumers’ ascription of responsibility is predicted by perceived behavioral control (H10).

A final set of predictions regarded the relations between trust and the other variables. We test whether trust in the natural food supply chain haa a positive effect on the intention to buy natural foods (H11). Finally, following past evidence showing that consumers’ trust predicts intentions related to food choices via the mediation of attitude [73], we hypothesize that consumers’ trust in the natural food supply chain has a positive direct impact on attitude (H12a) and an indirect effect on intention via attitude (H12b).

## 3. Materials & Methods

### 3.1. Participants and Procedure 

Then, our survey is on a probabilistic area sampling in Italy using a representative consumer panel. The sample include 1018 individuals responsible for food purchases within their households (mean age M = 41.09; SD = 12.56, ranging from 18 to 65 years old; female = 512; male = 506). To obtain a representative sample of the Italian population, we stratify consumers by gender, age, education, occupation, income class, geographical location, city/town size, and household members. The sample is highly educated (52.8% of consumers obtained a high school diploma and 32% obtained an university degree). Additionally, 45.8% of participants is married and the mean dimension of the family is of three members (M = 3.06; SD = 1.21). Most of the sample has a low monthly wage, ranging from EUR 1000 to 2000 (35.6%), or a medium monthly wage, ranging from EUR 2001 to 3000 (26.5%). Data collection was made by GfK, which is the fourth larger market research institute in the world. Participants were invited to provide written consent and to complete an online questionnaire. 

### 3.2. Measures

At the beginning of the survey, participants read a definition of natural food. We chose the one proposed by USDA, which, in our opinion, is the most exhaustive among the few existing: “A natural food is a food with no added artificial flavoring, and which has been minimally processed”. Then, they were invited to complete a questionnaire measuring consumers’ intention to purchase natural food products, TPB predictors (attitude, subjective norm, PBC), VBN predictors (general pro-environmental beliefs, awareness of consequences, ascription of responsibility, personal norm), and trust in the natural food supply chain. The scales of the questionnaire are adapted from past studies using the same study variables [50,74,75,76,77,78,79,80]. Each of the scales used concerning the purchase of natural food products is assessed on a 7-point Likert scale, ranging from ‘completely disagree’ (1) to ‘completely agree’ (7). Before collecting data with a national survey, we tested the face validity of the selected measurements by asking nine researchers in social psychology and economics to evaluate the degree to which each measure assessed the constructs that we intended to study. Then, we conducted a pre-test of the measurements with 88 undergraduate psychology or economics students. Given that the experts and the results of the pre-test confirmed the validity of our measurements, we did not make any adjustments to the questionnaire.

Table 1 shows standardized factor loadings for each item.

Intention to purchase natural food in the next month is measured using three items (e.g., “I intend to purchase natural food over the next month”; adapted from Armitage and Conner [50] Cronbach’s alfa is high (α = 0.94; composite reliability = 0.95).

Attitude towards purchasing natural food is measured with four items (e.g., “The purchase of natural food in the next month would be/is good”; Armitage and Conner, 1999) [82]. Cronbach’s alfa is high (α = 0.94; composite reliability = 0.96).

Subjective norm is assessed with four items (e.g., “Others believe that I should buy natural food”; adapted from Paul et al., 2016). Cronbach’s alfa is high (α = 0.88; composite reliability = 0.92).

Perceived behavioral control is assessed with three items (e.g., “I have the financial resources to buy natural food”; adapted from Lombardi et al., 2017). Cronbach’s alfa is satisfactory (α = 0.76; composite reliability = 0.86).

General pro-environmental beliefs are assessed with three items (e.g., “Human progress can only be achieved by maintaining an environmental balance”; adapted from López-Mosquera and Sánchez, 2012; Ojea and Loureiro, 2007; Stern et al., 1995). Cronbach’s alfa is high (α = 0.89; Composite reliability = 0.93).

Awareness of consequences in relation to purchasing natural food is assessed with three items (e.g., “Purchasing natural food improves our economy”; adapted from Teisl et al., 2009). Cronbach’s alfa is high (α = 0.88; Composite reliability = 0.93).

Ascription of responsibility towards purchasing natural food is assessed with three items (e.g., “Every citizen must take responsibility for the protection of the environment by purchasing natural food”; adapted from Gärling et al., 2003). Cronbach’s Alfa is high (α = 0.88; composite reliability = 0.92).

Personal norm towards purchasing natural food is assessed with three items (e.g., “I feel a moral obligation to purchase natural food for the protection of the planet”; adapted from Teisl et al., 2009). Cronbach’s alfa is high (α = 0.96; composite reliability = 0.96).

Trust in natural food supply chain is assessed with eight items (e.g., “I trust Italian institutions which certify natural foods”; adapted from Anisimova, 2016). Cronbach’s alfa is high (α = 0.96; composite reliability = 0.97).

### 3.3. Data Analysis

Our theoretical model has been tested using structural equation modeling (SEM) with MPLUS 7. To test the internal consistency among the observed variables of each latent variable, we use Cronbach’s alpha and composite reliability. We also test convergent and discriminant validities by analyzing factor loadings and AVE (average variance extracted) values. A full model is run to simultaneously include observed and latent variables in a mix of path analysis and confirmatory analysis. Specifically, a full model includes both a measurement model, which relates the variables to the constructs, and a structural model, which relates the constructs to other constructs [83]. 

The model fit is firstly estimated using a chi-square test. A non-significant chi-square test indicates that the model fits the data well [83]. However, the chi-square alone cannot verify the goodness of the model fit tested with large samples as the likelihood ratio test is too sensitive to sample size. A few other incremental goodness-of-fit indexes are used to address this problem [83]. The root mean square error of approximation (RMSEA) with a value of 0.05 or less indicates a good fit [84]. Cut-off values higher than 0.90 for the comparative fit index (CFI) and the Tucker–Lewis index (TLI) [85], as well as a value equal to, or less than 0.05 for the standardized root mean square residual (SRMR) [84], are generally considered to represent an acceptable fit.

## 4. Results 

The standardized item loadings of all study variables vary from 0.70 to 0.98 (Table 1), thus being highly significant. The high values of composite reliability in the study scales assure their internal consistency. The AVEs from latent variables range from 0.73 to 0.92, therefore they are above the recommended cut-off of 0.05 [86,87]. These results confirm that all observed variables present a high convergent validity. Discriminant validity is also attested by the results that all AVE values are higher than correlations between latent constructs [87]. Table 2 shows the means, correlations, and AVE values relevant to convergent and discriminant validity of the study variables.

All measures have a high mean, indicating that Italian consumers have positive beliefs vis-à-vis natural food purchases. Interestingly, the measure with highest mean and lowest reduced variance is general pro-environmental beliefs. This result indicates that Italian consumers generally perceive the importance of maintaining the environmental balance. Examination of the correlations (Table 2) indicates that positive attitude, awareness of consequences, ascription of responsibility, and personal norm in relation to buying natural food are the strongest correlates of Italians’ intention to purchase natural products.

The goodness-of-fit of the model is satisfactory (chi-square = 1349.029; df = 140; RMSEA = 0.05; CFI = 0.97; TLI = 0.97; SRMR = 0.03). Below we describe and report the standardized effects for each hypothesized path among the study variables (Figure 2).

As to the TPB variables, a positive attitude towards natural food (β = 0.47; *p* < 0.001) and perceived behavioral control (β = 0.31; *p* < 0.001) explains a high intention to purchase natural food, while this is not the case for subjective norm (β = 0.10; *p* = 0.18). These findings confirm our H1 and H2 but reject H3. 

As to the effects of the VBN variables, higher personal norm results in a greater intention to purchase natural food (β = 0.13; *p* < 0.001). Moreover, in line with the VBN model, personal norm is explained by ascription of responsibility in purchasing natural food (β = 0.99; *p* < 0.001), which in turn is explained by consumers’ awareness of consequences in relation to buying natural foods (β = 0.70; *p* < 0.001). Finally, consumer awareness of the consequences of purchasing natural food is significantly explained by general pro-environmental beliefs (β = 1.36; *p* < 0.001). These findings support H4, H5, H6, and H7a. 

Additional mediation analyses confirm the indirect effect of general pro-environmental beliefs on intention through awareness of consequences, ascription of responsibility and personal norm (Ind = 0.43; *p* < 0.001). This result confirms our hypothesis H7b.

As regards to the integration of the two models, we find that Italian consumers’ attitudes to buying natural foods are more pronounced when they have higher general pro-environmental beliefs (β = 0.12; *p* < 0.001), and especially when they are aware of the consequences of purchasing such food products (β = 0.68; *p* < 0.001), confirming both H8 and H9. Furthermore, as expected with H10, ascription of responsibility is higher when consumers have a higher perception of controlling their natural food purchases (β = 0.32; *p* < 0.001). 

Finally, trust in natural food supply chain explains intention both directly (β = 0.11; *p* < 0.001) and indirectly via attitude (β = 0.07; *p* < 0.001). Indeed, trust have a significant impact also on attitude β = 0.15; *p* < 0.001). Thus, hypotheses H11, H12a, and H12b are also supported.

Taken together, these results confirm our research hypotheses, showing that natural food purchasing can be effectively explained by an integrated model which combines a series of psychosocial antecedents including VBN dimensions, TPB dimensions, and trust. 

## 5. Discussion

The present study proposes and tests a psychosocial model to explain consumers’ intentions to purchase natural food products. The model integrates the TPB and the VBN theoretical frameworks. The TPB model has been selected to capture motives related to consumers’ decision-making process, such as positive or negative evaluations towards buying natural food and perceptions of having enough financial resources to purchase it. The VBN model has been selected to consider consumers’ pro-environmental motives, such as the moral feeling of purchasing natural food to protect the environment. In this integrated model, we also add the dimension of trust in the natural food supply chain. Most consumers are concerned about pesticides, food additives, and processed food and consumers expect institutions, food processing industries and retailers to take responsibility for food safety. Trusting these actors can therefore reduce feelings of uncertainty [88]. 

Our results are consistent with the proposed model and, as such, contribute to the advancement of literature regarding the psychosocial antecedents of natural food purchasing.

First, the present study suggests that participants’ intentions to purchase natural food is strongly associated with conscious determinants, as defined by TPB, as extensively demonstrated by past researches [32,89]. Attitude turns out to be the most important predictor of Italians’ intention to buy natural food, followed by perceived behavioral control. Previous research has already demonstrated that attitude and perceived behavioral control play a decisive role in the explanation of consumers’ food choice, such as the intention to buy organic food. However, in this study, we show, for the first time, the decision-making process when consumers decide to purchase natural food products. Considering the important role of attitude and perceived behavioral control as antecedents of the intention to purchase natural food, distributors might increase the purchase of this category of products by improving consumers’ knowledge about natural food and their perception of control. For example, both institutions and retailers may provide adequate information about how to recognize natural food at the market.

Second, our findings contribute to the literature about food choices by underling that rational motives are linked to moral motives [42]. Analyzing the role of VBN variables in predicting the intention to purchase natural food, we show that consumers’ personal norm is a relevant predictor. Besides, pro-environmental beliefs and the awareness of the environmental benefit of preferring natural food predict consumers’ attitude, showing that moral motives are precursors to rational considerations. This is important in light of the lack of consensus about the features embedded in natural products. Indeed, this category of products is valued by consumers for its supposed low impact on the environment. Therefore, some food features often paired with the attribute natural, such as the reduced pollution from their production or minimal packaging, are important to consumers. They represent distinctive characteristics that can be used to enhance the success of such products, and they can also be considered by policymakers in devising legislation to regulate the use of the term natural on food labels. 

Third, the present results contribute to the literature by emphasizing the importance of trust when purchasing natural food. Trust in the natural food supply chain is an important antecedent of purchase intention. This is in line with prior research according to which trust plays a relevant role, especially when consumers feel a lack of knowledge and familiarity regarding a purchase category, and therefore consider the actors of the supply chain as responsible for food safety and sustainability [66,90]. Our results suggest using trust in the natural food supply chain as leverage to attract consumers’ attention. For example, public markets and producers should favor the possibility to establish a direct relationship with the final consumer, increasing confidence in production processes. 

Then, based on the above findings, future research might study if our integrated model can be extended with other additional variables that could increase its predictiveness, such as food risk perception [91]. Food risk perception concerns the individual’s perception of the presence of a risky attribute in food, which can cause negative health consequences. Past studies have shown that food risk perception determines consumer attitudes [92], and in turn is predicted by consumers’ trust [91]. Thus, future studies may test if the path from trust to attitude, that we reported in this study, is mediated by food risk perception. Future research might also verify if subjective characteristics and socio-demographic factors influence the variables of the present model. For instance, previous scholars have shown that age, gender, and education influence the evaluation of food products, such as functional food [93] and local food [94]. Additionally, investigations in other countries may address whether there is a consumption pattern that varies geographically.

We are obliged to highlight that, despite consumers being motivated by moral motives to natural consumption, it is not possible to widely assess whether natural products may lower the environmental impact of food consumption, or improve the nutritional profile of the diet, due to the lack of a clear legal framework. The ultimate objective of this work is, then, a contribution in terms of raising awareness around what natural products are, and how their production takes place. This problem can be solved by policymakers who can work to define the legal framework surrounding this category of products. Therefore, they are invited to deal with the regulation of natural products.

Finally, the present research has some limitations that, in future, scholars might address. Although it was conducted on a representative sample of consumers, the sample was limited to Italy and results are therefore not generalizable to other populations. Besides, the knowledge around this topic would advance considering also what is the actual behavior of consumers, due to the widespread attitude behavior gap in consumers towards different products, especially the ones characterized by credence attributes. However, due to the different specifications of natural foods, these insights appear to be highly dependent upon the single case study chosen. Lastly, our model and related results offer interesting recommendations on how to design effective public communication to increase consumers’ attention towards natural food. 

## 6. Conclusions

In this study, after acknowledging the fragmented knowledge about natural food products and their missing legal framework, we investigated the psychological antecedents of the consumption of natural products. In detail, we analyzed the element driving the intention toward purchasing of food with *natural* labels. To this purpose, we developed an integrated model to explain consumers’ intentions. The aim is to contribute to an understanding of consumers’ motivations behind the purchase and appreciation of these products, and to raise awareness around this category of food, with the ultimate goal of fostering the process of building a legal framework that would define what the actual characteristics of the products are, behind the different perceptions that the consumers developed, regardless of the reality.

The study helped to understand what drives natural food intention in consumers. The first insight we obtained is relative to the importance of conscious determinants, as expected due to the established popularity of TPB in the food domain. In addition, rational motives are intertangled with moral motives, due to the health and environmentally friendly halo possessed by these products. A role is also played by trust, as happens for products characterized by credence attributes. 

Overall, the present study showed that the combination of TPB and VBN dimensions can provide more insightful explanations and explain the complexities of consumers’ intentions to purchase natural food. This model can be a starting point to develop further research on consumers’ perceptions and preferences for this more and more popular food category. Hopefully, such efforts might provide support for the development of an appropriate legal framework to regulate the use of *natural* as a food claim. 

## Figures and Tables

**Figure 1 nutrients-13-01904-f001:**
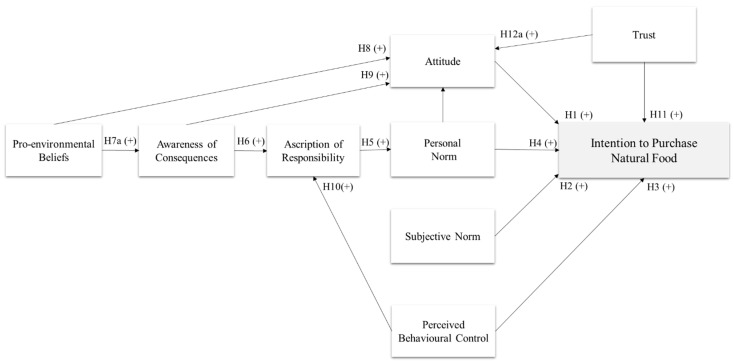
An integrated model to explain natural food purchases.

**Figure 2 nutrients-13-01904-f002:**
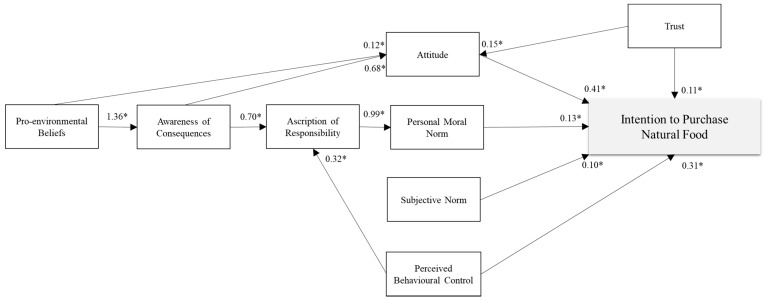
Standardized results of the proposed model. *Note.* * *p* < 0.01.

**Table 1 nutrients-13-01904-t001:** Measures used in the survey and related standardized factor loadings.

Measures	Standardized Factor Loading
Intention (adapted from [50])	
I intend to purchase natural food over the next month.	0.93
I plan to purchase natural food over the next month.	0.98
I will definitely purchase natural food over the next month.	0.88
Attitude (adapted from [50])	
The purchase of natural food in the next month would be/is good.	0.91
The purchase of natural food in the next month would be/is pleasant.	0.87
The purchase of natural food in the next month would be/is right.	0.91
The purchase of natural food in the next month would be/is useful.	0.89
Subjective norm (adapted from [80])	
Others believe that I should buy natural food.	0.80
Others would like me to buy natural food.	0.75
Most people purchase natural food.	0.80
The people I respect purchase natural food.	0.77
Perceived behavioral control (adapted from [76])	
I have the financial resources to buy natural food.	0.83
I have the opportunity to buy natural food.	0.73
I have the knowledge to choose and buy natural food.	0.84
General pro-environmental beliefs (adapted from [75,77,79])	
Human progress can only be achieved by maintaining an environmental balance.	0.70
Preserving nature now means securing the future of human beings.	0.92
Human beings can progress only by conserving natural resources.	0.89
Awareness of consequences in relation to buying natural food (adapted from [78])	
Purchasing natural food improves our economy.	0.70
Purchasing natural food improves our planet.	0.92
Purchasing natural food improves the environment.	0.89
Ascription of responsibility towards purchasing natural food (adapted from [74])	
Every citizen must take responsibility for the protection of the environment by purchasing natural food.	0.88
The authorities are responsible for the protection of the environment through the promotion of natural food purchase.	0.78
I’m responsible for the protection of the environment through the promotion of natural food purchase.	0.87
Personal norm (adapted from [78])	
I feel that I should purchase natural food for the safety of the environment.	0.88
I feel a moral obligation to purchase natural food for the protection of the planet.	0.86
I feel that I should buy natural rather than conventional food to protect the environment.	0.88
Trust in natural food supply chain (adapted from [81])	
I trust Italian institutions which certify natural foods.	0.83
I trust Italian natural food manufacturers.	0.90
I trust Italian natural food retailers.	0.91
I trust claims on natural food labels.	0.89
I trust the natural food products I buy (or that I could buy).	0.86
I can rely on natural food products sold in Italy.	0.89
I trust store personnel who sell natural foods.	0.83
I trust a product that carries a natural label/a natural certificate.	0.87

**Table 2 nutrients-13-01904-t002:** Means, correlations, and AVEs of the study variables.

Study Variables	1.	2.	3.	4.	5.	6.	7.	8.	9.	*M*	*SD*
1. Intention to Purchase Natural food	**0.88**									5.82	1.09
2. Attitude towards Natural Food Purchase	0.65	**0.85**								5.94	1.02
3. Subjective Norm	0.40 *	0.48 *	**0.74**							4.88	1.25
4. Perceived Behavioral Control	0.54 *	0.63 *	0.73 *	**0.73**						5.19	1.20
5. General Pro-environmental Beliefs	0.61 *	0.67 *	0.25 *	0.39 *	**0.83**					6.17	0.98
6. Awareness of Consequences	0.67 *	0.77 *	0.44 *	0.56 *	0.74 *	**0.81**				5.85	1.05
7. Ascription of Responsibility	0.64 *	0.80 *	0.52 *	0.63 *	0.66 *	0.79 *	**0.92**			5.73	1.09
8. Personal Norm	0.63 *	0.79 *	0.55 *	0.68 *	0.61 *	0.80 *	0.90 *	**0.90**		5.64	1.15
9. Trust in Natural Food Supply Chain	0.33 *	0.40 *	0.50 *	0.55 *	0.23 *	0.37 *	0.40 *	0.43 *	**0.79**	5.09	1.23

The bold values in the diagonal row are the AVE values of the latent variables. The numbers below diagonal are the correlation coefficients between the latent variables. * *p* < 0.001. AVE = average variance extracted.

## Data Availability

The data presented in this study are available on request from the corresponding author.

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
