# Peer review of "Why Do Consumers Intend to Purchase Natural Food? Integrating Theory of Planned Behavior, Value-Belief-Norm Theory, and Trust"

_nutrients, 2021, doi:10.3390/nu13061904_

Round 1

Reviewer 1 Report

Title: Why Do Consumer Intend to Purchase Natural Food? Integrating Theory of Planned Behavior, Value-Belief-Norm Theory, and Trust This paper aims at leveraging two well-established theories, namely theory of planned behavior and value-belief-norm theory, to explain consumer intend to purchase natural food. The authors conducted a quantitative study to investigate this phenomenon. In general, the paper, as is, lacks sufficient depth to make a significant contribution. My major concerns followed by minor issues are presented below. 1. On lines 41-46, the authors mentioned “Natural labels can also overlap in their meaning with other claims, such as organic and the so-called clean labels: They can be “free from” labels of reformulated products with the absence of ingredients that are perceived to be somewhat harmful [9]. As a matter of fact, the organic labelling is strictly normed. So, there may be a confusion in the perception of consumers but formally there are two distinct and separated labels for natural and clean food (Regulation (EC) n. 834/2007).” The authors need to distinguish between natural food, organic food, and clean food in a clear and concise manner. 2. The Introduction is not compelling. What is the difference/ similarity characteristics between these foods (i.e., natural food, organic food, and clean food)? Because the authors have suggested similarity characteristics between these foods, the authors need to strengthen the entire paper demonstrating better justifications concerning the importance of this research - specifically on areas that are new and not something that has been "repackaged" with limited groups of consumer, say natural food consumers. The Introduction is not compelling and not clearly state the contribution to theory and why this is important. 3. The authors have cited Chen's study (2020). Chen (2020) has combined the VBN and TPB models to explain people's consumption intention of locally produced organic foods. If there is a similarity in natural food and organic food, then the authors need to clearly show how their paper builds on that to contribute to the literature. What is new here? The authors need to clearly describe upfront the findings of Chen (2020) and clearly show how their paper builds on that to contribute to the literature 4. The authors provide the following definition of natural food: “A natural food is a food with no added artificial flavoring, and which has been minimally processed.” The authors do not convincingly argue that natural foods are actually contributing to the environment and animal welfare. So that raises the question why it is important to study natural food from moral norms perspective. 5. Another important concern derives from misunderstandings and inaccuracy: some papers do not claim the fact or do not contain the information that has been referred in the manuscript (e.g., Reference #12 on line 67; Reference #14 on line 67). It can raise the further problem of research motivation. 6. The sources of the used scales are not provided. Please explain to us regarding the scales come from and why the authors chose these scales rather than others. 7. The items used to measure the research constructs do not appear to be appropriate. For example, what facet of perceived behavioral control is captured by “I have the possibility to buy natural food”? The item appears to tap into “intent” rather than “ability”. 8. On table 1, the author labels that the trust are " Trust in natural food". Looking at the items used, this seems to be a quite different construct. Was the authors questionnaire pre-tested for face and content validities? 9. The tests of convergent and discriminant validity are very weak tests. I see it as an important issue here that the authors clearly show that their research constructs are in fact distinct and discriminant. At present the authors’ current tests are very weak ones and don’t leave the reader with a lot of confidence as to their discriminant and convergent validity. 10. Need to provide other goodness of fit indicators (CFI, RMSEA etc.), it is essential that you show all fit statistics. At present, I can’t find these anywhere in the manuscript. 11. Please clarify the mediating effect analysis. 12. Discussion section should contain more cited references (i.e. previous researches in relevant areas) to describe the significance of this research. Implications of the major findings of this research should be derived by the analysis of more previous researches.

Author Response

1. On lines 41-46, the authors mentioned “Natural labels can also overlap in their meaning with other claims, such as organic and the so-called clean labels: They can be “free from” labels of reformulated products with the absence of ingredients that are perceived to be somewhat harmful [9]. As a matter of fact, the organic labelling is strictly normed. So, there may be a confusion in the perception of consumers but formally there are two distinct and separated labels for natural and clean food (Regulation (EC) n. 834/2007).” The authors need to distinguish between natural food, organic food, and clean food in a clear and concise manner.

We are sorry if the aims of the article are not clear, to this purpose we defined more clearly the category of products that we are investigating in lines 43-48

2. The Introduction is not compelling. What is the difference/ similarity characteristics between these foods (i.e., natural food, organic food, and clean food)? Because the authors have suggested similarity characteristics between these foods, the authors need to strengthen the entire paper demonstrating better justifications concerning the importance of this research - specifically on areas that are new and not something that has been "repackaged" with limited groups of consumer, say natural food consumers. The Introduction is not compelling and not clearly state the contribution to theory and why this is important.

Thank you for rasing this point, the core of the research deals to identify the drivers for consumption of a category of products that actually exists on the market and is very popular. Although it has never been regulated or identified clearly. The perception of consumers may lead to overlap some labels, so we dealt with this confusion. But this study is strictly aimed at investigating the antecedents of purchasing of products that have the label containing the word “natural”. Organic certified and “free-from” labelled products have been excluded from this study. As specified in lines 43-48.

3. The authors have cited Chen's study (2020). Chen (2020) has combined the VBN and TPB models to explain people's consumption intention of locally produced organic foods. If there is a similarity in natural food and organic food, then the authors need to clearly show how their paper builds on that to contribute to the literature. What is new here? The authors need to clearly describe upfront the findings of Chen (2020) and clearly show how their paper builds on that to contribute to the literature

We thank you for this remark. We have now better explained Chen’s contribution and clarified that we contribute to the literature by extending the integration of the VBN and TPB models with the additional role of trust in natural food supply chain.  See lines 103-114.

4. The authors provide the following definition of natural food: “A natural food is a food with no added artificial flavoring, and which has been minimally processed.” The authors do not convincingly argue that natural foods are actually contributing to the environment and animal welfare. So that raises the question why it is important to study natural food from moral norms perspective.

One of the final objectives of the paper is to raise awareness around the natural label. It is actually present on the market and valued both by producers and by consumers. Despite the actual characteristics of the products are unknown, consumers are motivated to consumption by moral motives. It is beyond our scope to understand whether these products actually contributes to human welfare. However, raising awareness can lead to a regulation for this label, it may indeed lead to a definition of what can exactly be or not be natural also in terms of environmental impact.

So far only organic products can be claimed to be produced with a reduced environmental impact. This has been specified in lines 408-415

5. Another important concern derives from misunderstandings and inaccuracy: some papers do not claim the fact or do not contain the information that has been referred in the manuscript (e.g., Reference #12 on line 67; Reference #14 on line 67). It can raise the further problem of research motivation.

We are sorry for the mismatch of references, actually the #12 has been deleted, while the #14 is a review that highlighted all the groundless perceptions of consumers towards natural products

6. The sources of the used scales are not provided. Please explain to us regarding the scales come from and why the authors chose these scales rather than others.

All the references were reported in the main text, we have now included them also in Table 1.

7. The items used to measure the research constructs do not appear to be appropriate. For example, what facet of perceived behavioral control is captured by “I have the possibility to buy natural food”? The item appears to tap into “intent” rather than “ability”.

We thank you to allow us to clarify this point. We selected the measure from the literature review and all the items were adapted from recognized studies in the psycho-social domain, as reported in the Method section (p. 6-7). Regarding the measurement of perceived behavioural control, we have now correctly translated the Italian word “possibilità” in “opportunity” instead of “possibility”. Accordingly to the literature on the TPB, we actually operationalized the PBC as the belief that one has external and internal resources and opportunities to perform the behavior (in our case, “having financial resources to buy natural food”, “having knowledge to choose natural food”, “having the opportunity to buy natural food”).

8. On table 1, the author labels that the trust are "Trust in natural food". Looking at the items used, this seems to be a quite different construct. Was the authors questionnaire pre-tested for face and content validities?

We thank you for this precious comment. We have now better labeled this construct as “trust in natural food supply chain”. We have now added information about face validity and pre-test in the Methods (p. 5).

9. The tests of convergent and discriminant validity are very weak tests. I see it as an important issue here that the authors clearly show that their research constructs are in fact distinct and discriminant. At present the authors’ current tests are very weak ones and don’t leave the reader with a lot of confidence as to their discriminant and convergent validity.

We have now included analyses to test the internal consistency, convergent validity and discrimant validity. (p. 7, 270-273; p. 7, 238-268; p. 8,lines 287-295 and Table 1).

                        10. Need to provide other goodness of fit indicators (CFI, RMSEA etc.), it is essential that you show all fit statistics. At present, I can’t find these anywhere in the manuscript.

In our analysis, we had already reported Chi-square, RMSEA , CFI, TLI and SRMR (p. 8, lines 310-312).

                              11. Please clarify the mediating effect analysis.

We have now clarified the mediating effect analysis (p. 9, 327-329).

12. Discussion section should contain more cited references (i.e. previous researches in relevant areas) to describe the significance of this research. Implications of the major findings of this research should be derived by the analysis of more previous researches.

Thank you for raising this point, some additional references has been inserted in the discussion part to link our findings in the big picture of past research.

Reviewer 2 Report

This paper presents an interesting study based on a theoretical model that connects Theory of Planned Behavior, Value-Belief-Norm Theory and Trust of consumers to natural food. The general aim of this study was to explain consumers' intention to buy natural food by verifying the predicting role of conscious decisions and moral motives.

The presented manuscript is interesting and fills a research gap in the literature.  A range of information has been collected and presented in an overview that addresses the stated goal.

However, some comments and recommendations can be made - pdf.

Author Response

Reviewer 2

1/ The abstract is formulated correctly. However, there are no calculation details and the results are presented in graphs, which in my opinion should be considered sufficient.

We thank you for your positive feedback.

2/ The authors did not consider to what extent the sample reflects the population. That is, N is high (N=1018) but to what extent did the demographic, economic and social variables correspond to the population?
The authors write about a representative sample - the article has no scientific evidence in this regard. Please provide a thorough explanation in this regard:

- What was the sampling method used?
- How was the data collection conducted?
- How was the survey questionnaire constructed?
- Who was consulted for the survey questionnaire?
- What criteria were used to select survey participants?
- How was the survey questionnaire pre-tested?
- What adjustments were made to the final version of the survey questionnaire after pre- testing?
I would like to point out the following facts - Does the sample structure of a country represent the population structure of that country? With this sample this was possible. For example, what were the proportions of gender, age groups? Did the authors achieve over- representation of any group (e.g. women, older, younger, etc.) in the survey conducted? An imbalance of proportions e.g.: gender makes it difficult to draw conclusions.

Thank you for bringing this to our attention.

As stated in the Participants and Procedure section, we used a probabilistic area sampling method.

Data collection was conducted by the GfK, one of the leading market research organizations in the world.

We have now clarified how the questionnaire was constructed and how we tested face validity (p. 7,  228-234).

The inclusion criterion was being responsible for food purchases within households.

We have now specified how we pre-test the questionnaire and its related adjustment (p. 6, 231-235).

To obtain a representative sample of the Italian population, the agency that recruited participants stratified consumer by  the following criteria: education, occupation, income class, geographical location, city/town size, number of household members, gender and age (line 207-209). Thus, there was not an imbalance of proportion in any of these criteria.

3/ I suggest that the authors in the discussion also refer to other variables that would be worth extending the model to in the future.The TPB model is often extended but this element needs to be justified in the literature or verified empirically.
The TPB model is often extended (extended model of TPB) but this element needs to be justified in the literature or verified empirically.

We thank you for raising this point. In the discussion, we have now referred to other variables that would increase the predictiveness of our model (lines 395-407).

4/ I suggest supplementing the discussion part, it does not exhaust the current state of knowledge.

Thank you for the suggestion, some references have been added with the aim of specifiyng the context in which our findings rely, considering past literature. Some additional insights have been added in lines 408-423

5/ The conclusions are very short.I strongly recommend expanding and completing this part. For example, additions regarding legislation and statements, marketing activities of companies. The research is interesting but its possible application is not sufficiently

described. Supplementation of the discussion, implications will certainly increase the scientific quality of this article.

Thank you for this suggestion, accordingly, the conclusion part has been integrated. See lines 429-444.

6/ Do the authors have any other recommendations for further research that could be prepared based on the presented results?
This work could be treated as a study presenting the construction of the research tool, thus making it possible to repeat the study, for example, in other countries or to check the results in the same country. In this case I would suggest to the authors to make the questionnaire an annex.

We have now extended our discussion with recommendations for further research in more detail and added suggestions for policymakers (lines 402-423)

7/ Please explain the limitations of the study.

The limitations of the study have been presented more clearly in lines 416-426

Round 2

Reviewer 1 Report

The authors have sufficiently tried to answer my comments.

Reviewer 2 Report

Dear Authors,
thank you for considering my suggestions and I appreciate your response. After reviewing the content of the new Nutrients-1221134 manuscript "Why Do Consumers Intend to Purchase Natural Food? Integrating Theory of Planned Behavior, Value-Belief-Norm Theory, and Trust" I would like to conclude that the issue addressed is in line with the theme of the journal. The problem presented in the article is important and interesting, the research question was formulated clearly enough.
